# Performance Test of the Rotational Sensor blueSeis-3A in a Huddle Test in Fürstenfeldbruck

**DOI:** 10.3390/s21093170

**Published:** 2021-05-03

**Authors:** Gizem Izgi, Eva P. S. Eibl, Stefanie Donner, Felix Bernauer

**Affiliations:** 1Institute of Geosciences, University of Potsdam, Karl-Liebknecht-Str. 24-25, 14476 Potsdam-Golm, Germany; evapseibl@uni-potsdam.de; 2Federal Institute for Geosciences and Natural Resources (BGR), Stilleweg 2, 30655 Hannover, Germany; stefanie.donner@bgr.de; 3Department of Earth and Environmental Sciences, Ludwig Maximilian University Munich, Theresienstr. 41, D-80333 Munich, Germany; fbernauer@geophysik.uni-muenchen.de

**Keywords:** rotational seismology, huddle test, coherency, source direction, coherent noise, blueSeis-3A sensors

## Abstract

Rotational motions play a key role in measuring seismic wavefield properties. Using newly developed portable rotational instruments, it is now possible to directly measure rotational motions in a broad frequency range. Here, we investigated the instrumental self-noise and data quality in a huddle test in Fürstenfeldbruck, Germany, in August 2019. We compare the data from six rotational and three translational sensors. We studied the recorded signals using correlation, coherence analysis, and probabilistic power spectral densities. We sorted the coherent noise into five groups with respect to the similarities in frequency content and shape of the signals. These coherent noises were most likely caused by electrical devices, the dehumidifier system in the building, humans, and natural sources such as wind. We calculated self-noise levels through probabilistic power spectral densities and by applying the Sleeman method, a three-sensor method. Our results from both methods indicate that self-noise levels are stable between 0.5 and 40 Hz. Furthermore, we recorded the 29 August 2019 $M_{L}$ 3.4 Dettingen earthquake. The calculated source directions are found to be realistic for all sensors in comparison to the real back azimuth. We conclude that the five tested blueSeis-3A rotational sensors, when compared with respect to coherent noise, self-noise, and source direction, provide reliable and consistent results. Hence, field experiments with single rotational sensors can be undertaken.

## 1. Introduction

Three components of translation, three components of rotation, and six components of strain define the complete deformation field caused by an infinitesimal deformation under the assumption of classical elasticity [1]. Translation and rotation hereby describe the complete seismic wavefield. Strain and translation of the total motion can be described using strain meters and classical seismometer recordings, which provide translational ground displacement. However, to fully understand the ground motion, recording three components of rotation is essential. Although the rotational part is considered useful in seismology, due to the lack of appropriate instrumentation, it could not be measured directly until recent years [2].

To directly measure rotational ground motion, one can use ring-lasers; rotational sensors; or indirectly, the method of array-derived rotation (ADR) [3]. Fiber-optic gyroscopes (FOGs) are one such rotational sensor and measure rotation by exploiting Sagnac’s effect. When considering a case of ideal circular path, light entering the interferometer splits into two counter-propagating waves and travels the same path in phase. However, a phase shift occurs if the coil rotates. Measuring this phase shift allows for estimating the rotation rate [4].

Fiber-optic gyroscopes have started to be commonly used in the field of seismology (e.g., [5,6,7,8,9]). Portable blueSeis-3A sensors (iXblue, France), which are one of the first FOGs that only serve seismological purposes, allow for measuring three components of rotational motion with a high sensitivity in the broad frequency range from 0.001 to 50 Hz. It is essential to investigate the data quality and consistency of recorded signals between newly developed instruments. Thus far, Bernauer et al. [5] performed laboratory experiments to test these sensors. After testing the susceptibility of the equipment for varying temperature and magnetic field conditions, they stated that blueSeis-3A data are reliable for various field applications such as volcanology, ocean-bottom studies, seismology, and earthquake engineering. Another study that discusses the reliability of rotational data from blueSeis-3A is Yuan et al. [9]. They compared the rotational data derived from ADR with rotational sensor recordings of the 2018 $M_{w}$ 5.4 Hualien earthquake in eastern Taiwan and the earthquake sequence from $M_{L}$ 3.3 to $M_{w}$ 4.1 2016 in central Italy. They concluded that the back azimuth results from rotational sensors are consistent with the cross-correlation and polarization analysis applied.

The whole seismic recording of a station contains coherent seismic signals, incoherent seismic noise, and the self-noise of the instrument. This instrumental self-noise may dominate when the instruments are deployed in quiet and remote sites. This is particularly the case for frequencies below 1 Hz [10]. Measuring instrumental noise is essential because we need a good estimate of the self-noise level so that we can eliminate time windows where self-noise dominates. For the coherency analysis, we applied the single- [11] and three-instrument [12] method.

In this paper, we present the results from a passive field experiment that includes six rotational and three translational sensors. Our scope is to analyze the performance of the rotational sensors with respect to each other and in comparison with seismometers. To investigate the performance of multiple blueSeis-3A sensors in combination with translational seismometers, a huddle test was performed at the Geophysical Observatory in Fürstenfeldbruck, Germany, as a collaborative work of Ludwig-Maximilians University of Munich (LMU), University of Potsdam (UP), and the Federal Institute for Geosciences and Natural Resources (BGR). In the following, we explain the instrumental setup (Section 2) and present our methodology to investigate the self-noise and the source direction in Section 3. We discuss the instrumental noise with two different methods in Section 4.1.1. In Section 4.2.1, we discuss the quality and reliability of the data over coherent noise and correlation study. In Section 4.3, we present our results from coherency analysis and back azimuth calculations with multiple sensors on the Dettingen earthquake.

## 2. Instrumental Setup

We installed six blueSeis-3A rotational sensors and three Trillium Horizon 120s seismometers (Nanometrics, Kanata, ON, Canada) on the basement floor of the archive building at the Observatory Fürstenfeldbruck, Germany (network X3). However, one of the rotational sensors did not produce data (XB103) and the data from one of the translational sensors (HRZ2) could not be used due to a technical error. The distance between all sensors was about 35 to 40 cm, which we assumed is sufficiently close for a co-located measurement. A sketch of the experimental setup that shows the relative positions of the instruments is given in Figure 1, while the recording times and positions are given in Table 1.

The sensors were first levelled and then aligned to geographical north using two Quadrans, one provided by UP (QUP) and one by LMU (QLMU). Quadrans are fiber-optic gyro-compasses that allow for a true north alignment in seismological experiments. We measured for about 20 min with both Quadrans to ensure stable direction measurements. We noticed a small misalignment of 0.6${}^{{}^{\circ}}$ to 1.2${}^{{}^{\circ}}$ between QUP and QLMU. While the XB103 sensor did not work from the beginning, the XBLUE sensor stopped recording too early and the data collected from the HRZ2 seismometer were unusable. The data were recorded at a 200 Hz sampling rate.

## 3. Methodology

The rotational and translational data were read, and time-shifts caused by the different time synchronization, time reference clocks, and low-pass filtering prior to digital sampling decimation were applied. The applied time-shifts, which maximize the correlation between vertical rotation rate and transverse acceleration, were −0.03 s for XB101, XB102, BS2, and XBLUE and −0.005 s for BS1, which are consistent with the values reported by Bernauer et al. [6]. Subsequently, these data were merged, detrended, tapered, and filtered between 1 and 50 Hz. Additionally, we applied instrumental correction to the translational data recorded by the seismometers. The software toolbox ObsPy [13,14,15] Version: 1.2.1 and Pyrocko [16] v2020.10.26 were used for data analysis.

### 3.1. Coherence

To characterize the self-noise of rotational sensors, we applied single- and a three-instrument methods. The single-instrument method is a simple calculation of probabilistic power spectral densities of self-noise and background noise, according to Holcomb et al. [11], applying direct Fourier transforms on the data, which might not reflect the true self-noise in high-noise environments. Sleeman et al. [12] developed a method that has become widely established for determining self-noise with three instruments, which provides more accurate results by calculating the statistical spread of the noise. These two methods are commonly used for traditional seismometers based on the electromechanical principles of mass and damping. Here, the instruments are based on the principle of fibre-optic gyroscopes and we are convinced that these methods are also applicable for rotational data [5,17].

The coherence $\gamma_{ij}^{2}$ between instruments i and j is given by [18];
(1)$$\gamma_{ij}^{2}=\frac{|P_{ij}{|}^{2}}{P_{ii}P_{jj}}$$
where *P* is the cross power of the denoted instruments. The cross power between two instruments denoted by *i* and *j* can be calculated as follows:(2)$$P_{ij}=H_{i}X_{i}\bar{H_{j}X_{j}}$$
where $H_{i}$ is the Fourier transform of the impulse response, $X_{i}$ is the Fourier transform of the input signal, and the bar denotes the complex conjugate of the Fourier transform of the corresponding sensors.

The single instrument method follows the simple relation to obtain coherency:(3)$$N_{ii}=\frac{P_{ii}}{H_{i}\bar{H_{i}}}.$$

The equation holds under the assumption of $X_{i}<<N_{ii}$, where $N_{ii}$ corresponds to the sensors’ self-noise. Even if this assumption is not satisfied in all frequencies, it is essential and often possible to estimate self-noise in a specific frequency band. For further information, the reader is referred to McNamara & Buland [19]. They used time windows of 1 h in length with an overlap of 50% to calculate the PPSDs. However, they also stated that, for frequencies higher than 20 Hz, shorter time windows should be analyzed.

Another technique for coherency analysis is the three instrument method, which is more robust since it is insensitive with respect to the errors in the transfer functions (e.g., [20]). This method is defined by the following equation that holds by assuming that the three sensors recorded the same input signal:(4)$$N_{ii}=\frac{P_{ii}-P_{ij}\frac{P_{ik}}{P_{jk}}}{H_{i}\bar{H_{j}}}$$
where $\frac{P_{ik}}{P_{jk}}$ are the relative transfer function between two instruments (*i* and *j*). For further comparison between these two methods, the reader is referred to Holcomb et al. [11], Sleeman et al. [12], Ringler et al. [18].

### 3.2. Direction Calculation

Rotational observations can be used to determine the back azimuth (BAz) of the source and seismic phase velocity information despite being a point measurement. We applied the method suggested by Hadziioannou et al. [21] to estimate the direction of the coherent seismic signals. They used cross-correlation coefficients to calculate the maximum correlation between vertical rotation rate ($\dot{\omega_{z}}$) and transverse acceleration ($a_{T}$) waveforms.
The transverse acceleration is given by the following:(5)$$a_{T}=-k^{2}c^{2}A\sin\left(kx-kct\right)$$
The vertical rotation rate is given by the following:(6)$$\dot{\omega_{\zeta }}=\frac{1}{2}k^{2}cA\sin\left(kx-kct\right)$$
From the transverse acceleration and vertical rotation rate follows
(7)$$\frac{a_{T}}{\dot{\omega_{z}}}=\frac{-k^{2}c^{2}A\sin\left(kx-kct\right)}{\frac{1}{2}k^{2}cA\sin\left(kx-kct\right)}=-2c$$
where *A*, *k*, *c*, and ω are the amplitude, wave number, phase velocity, and vertical rotation rate, respectively. Equation (Equation 7) shows that, for the co-located translational and rotational measurements, the phase velocities can be estimated from the amplitude ratios. To obtain the transverse component of acceleration, the north and east components need to be rotated around the (unknown) BAz. Therefore, in a specified sliding time window and in a loop over all possible BAz, the correlation of the resulting transverse component of acceleration with the vertical rotation rate is calculated. The BAz providing the highest correlation within the sliding window is assumed to be the correct BAz.

### 3.3. Allan Deviation

One way of characterizing the performance of fiber-optic gyroscopes is to calculate the so-called Allan deviation [22]. Despite being new in the field of seismology, the Allan deviation is commonly used in inertial navigation field. Self-noise and the drift of the instruments can be provided in terms of Allan deviation [5]. The Allan deviation is a time-domain measure and describes the stability of the sensor readout after averaging over a certain time span τ. Short-term noise of a sensor in terms of $nrad/s/\sqrt{Hz}$ can be found using the numerical correspondence of the Allan deviation at τ=1s. The Allan deviation σ(τ) is a function of the averaging time τ and is calculated as follows:(8)$$\sigma^{2}\left(\tau \right)=\langle \frac{{\left({\bar{y}}_{k+1}\left(\tau \right)-{\bar{y}}_{k}\left(\tau \right)\right)}^{2}}{2}\rangle ,$$
with ${\bar{y}}_{k}\left(\tau \right)$ as the *k*th average value of the time series *y* of length τ and 〈〉 denoting the average over all k along the time series *y*. Additionally, the type of noise is indicated by the slope of the Allan deviation. For instance, a −1/2 slope corresponds to the white noise domination over self-noise in that period range [5]. For more information about the Allan deviation, the reader is referred to Lefevre [4].

## 4. Results and Discussion

### 4.1. Self-Noise Characterization for Each Rotational Sensor

#### 4.1.1. Probabilistic Power Spectral Densities (Single-Instrument Method)

In general, this method does not require screening the data for a quiet period before calculating the PPSD. Thus, it is possible to both evaluate the station quality and the level of site noise. However, because we are interested in the self-noise, we inspected the waveforms and selected two days without disturbances in the vicinity of the station setup. These days were Thursday, 29 August (doy 241, Figure 2) and Sunday, 1st September (doy 244, Figure 3).

Here, we used the data from the entire day split in time windows of 10 min in length with an overlap of 75%. In Figure 2 and Figure 3, components HJ1 (**a**),(**d**),(**g**),(**j**),(**m**) and HJ2 (**b**),(**e**),(**h**),(**k**),(**n**) correspond to the east and north components while component HJ3 (**c**),(**f**),(**i**),(**l**),(**o**) corresponds to the vertical component (measuring torsion), respectively (measuring tilt).

In general, all five rotational sensors show stable self-noise power spectra below 50 Hz and perform equally well. Above 50 Hz, a Butterworth filter with a corner frequency at 70 Hz and a suppression of ∼70 dB, where $\mathrm{dB}=\;\mathrm{nrad}/s/\sqrt{\mathrm{Hz}}$ at Nyquist frequency, is implemented as anti-alias filter. The sensors BS1 and BS2 show increased amplitudes between 40 and 50 Hz. This is most likely because BS1 and BS2 are the first rotational instruments developed by iXblue, which at the time of the study required further improvements. We observe higher self-noise levels below ∼0.2 Hz. Another distinctive feature is the small amplitude increase at ∼10 Hz. This feature was already visible in Bernauer et al. [5], in their Figure 5, although it used different rotational sensors at a different testing site at laboratory conditions. In our data, it is visible on all weekdays but not on the weekends. Because of these two reasons, we assume that it is related to human activity.

#### 4.1.2. Power Spectral Densities with the Three-Station Method

This method was first introduced by Sleeman et al. [12] to estimate the self-noise levels of translational seismometers. Bernauer et al. [17] and Bernauer et al. [5] then adapted the method for the use for rotational sensors. In Figure 4, changes in the rotation rate of five rotational sensors and three components (East, North, and vertical) with respect to frequency are presented.

In general, all sensors show stable results between 0.5 to 40 Hz. Above 50 Hz, the self-noise of the sensors increased. We speculate that this increase is caused by inappropriate anti-alias filtering implemented on sensors. For the east component of the BS2 sensor and for the north component of the BS1 sensor, the PSD results are the most distorted. In contrast, they both showed higher PSD results on the vertical components above 50 Hz. Furthermore, the self-noise of XB102 and XB101 are more stable for the east and north components. Here, we also observe the small increase in the self-noises at 10 Hz but not so distinctively for the vertical component. However, this feature is not visible on recordings from the XBLUE sensor. Since the 10 Hz peak is visible on all sensors when applying the single station method but not when applying the three-station method, we argue that the 10 Hz peak is a coherent contribution from ambient seismic noise.

We observe a stable decrease in Allan deviations in the averaging time () between 0.04 to 90 s for all sensors and three components (Figure 5). For BS1 and BS2 and between 90 s and 110 s integration times, the Allan deviation reaches its minimum at 1.5 nrad/s to 3.0 nrad/s. At one second averaging time, the sensors show an Allan deviation between 15 nrad/s and 25 nrad/s. This is in good agreement with the nominal specifications that guarantee a sensor self-noise level of 25 nrad/s/$\sqrt{\mathrm{Hz}}$ from 0.01 Hz to 50 Hz and is consistent with the findings of Bernauer et al. [5]. The slope of the Allan deviation is approximately −0.5 in a frequency range from 0.05 Hz to 50 Hz. This value corresponds to the main contribution of white noise in that frequency range and is consistent with the flat PSD shown in Figure 4.

### 4.2. Recordings of Coherent Noise Groups and Incoherent Noise

#### 4.2.1. Coherent Noise Spectrograms

Following the instrumental self-noise analysis, we created catalogs of coherent noise signals. We found about 119 coherent signals in the entire week, which were visible on the rotational sensors. These events are visible neither in the recordings from the permanent seismometer FUR nor in the ring laser ROMY in Fürstenfeldbruck at about 250 m distance. We sorted them in five groups according to the duration, frequency content, and general shape of the signal. To better illustrate the spectral content of each group, we tested different window lengths and overlaps and present the spectrograms with optimal values here. Events within each group are visible on all sensors. Events in three out of five groups are visible on all components of each sensor. Events in the other two groups are not visible on the vertical components of all sensors except BS1 and BS2. To investigate the relative sensitivity of the co-located sensors, we plotted the coherent noise spectrograms of all sensors. In this section, we present one example from each group for all three components: HJ1 (East), HJ2 (North), and HJ3 (vertical) of five rotational sensors.


**(1) Long-Lasting Noise**


We observed an harmonic noise lasting approximately three hours with a fundamental frequency of 48 Hz (Figure 6). In the spectrogram, we used a 4 s window length with an overlap of 70%. This harmonic noise is visible every day between 06:00–09:00, 15:00–17:30, and 22:00–01:00 on the east and north components of both rotational and translational instruments. However, on the vertical component, the long-lasting noise is only recorded by BS1 and BS2.

The beginning and end of this harmonic noise are sharp and instantaneous. Hence, we rule out a machine as the possible source because the change in the spectral densities should be visible for a few seconds when that machine starts working or shuts down. As stated in Bormann et al. [23], power plants and wind turbines create a narrow-band noise around 50–60 Hz and sub-related harmonics at 25–30 Hz. Although we do not observe sub-related harmonics, the resemblance of the shape, frequency content, and the occurrence times might indicate that the noise was created by one of the power plants near Munich. However, since the power plants have a large distance to our site and we cannot detect cars on the nearby road, we think that a power plant is not the source [24]. Instead, we suggest that this noise is generated by an electrical device that creates a noise around 50 Hz which repeats every day in the same three time windows. Since the occurrence times for this noise is consistent with the scheduled dehumidifier system in the room, we suggest that this harmonic noise is caused by the electrical coupling of the dehumidifier system.


**(2) Emergent Burst-like Noise-Lasting Noise**


We observed eight emergent burst-like events with broad frequency content (10–50 Hz) and a duration of around 20 s (Figure 7). They were visible on both translational and rotational sensors. To better present the emergent burst-like events, we chose a 3 s window length with 70% overlap in the spectrogram.

All burst-like coherent noise events in our data set have an emergent onset and a slowly decaying coda. They are randomly distributed in the data set, which makes it challenging to identify the source. In Figure 7, we observe a pulse followed by an emergent burst-like event. Spectral densities of the pulse are dominantly visible on the vertical component of the BS1 and BS2 sensors. We observe higher spectral density values in Figure 7 from 30 to 50 s. Because the spectral density amplitudes are quite high, we assume that the source is either very close or very energetic. However, we could not identify it.


**(3) High Frequency Spike Shaped Noise**


We observed high-frequency spike events in the 30–50 Hz frequency band (Figure 8). We used a 3 s moving time window with an overlap of 70%. There are four events in this group and the noise repeats for approximately two minutes while the duration of each spike ranges from 0.2 to 0.5 s. All examples occurred on 29 August, Thursday, in the morning from 09:30 to 12:30 am local time.

The high-frequency spike-shaped noise is visible on all components of all sensors and can be associated with one of the coauthors working in the basement during a maintenance check of the instruments on Thursday morning.


**(4) Very Short Noise**


We observed very short events with a total duration of less than 4 s, whose waveform shape is similar to earthquakes (Figure 9). The first part of the event has a frequency range of 40–50 Hz, whereas the second part is similar to an earthquake coda with a broader frequency range (10–50 Hz). Events in this group are visible on all components of each sensor. Here, we used moving time windows of 0.4 s lengths with an overlap of 90%.

We observed five examples within this class although they are very small. No pattern in their timing throughout the day or week was detected. Auweraer et al. [25] stated that slamming a door is characterized by its extremely short duration and that both the materials of the door and its counterpart may create two different impacts visible in the spectrogram. When a door is closed, it may create oscillations. We suggest that this event group is caused by slamming doors and that the materials of the door and counterparts cause primary and secondary impacts in the waveforms.


**(5) Spike Shaped Noise in Broad Frequency Range (10–50 Hz)**


In our data set, we observed spike-shaped noise in the frequency range between 10 to 50 Hz (Figure 10). These events are the most common coherent noises that occurred every morning from 5 am until noon. We used a 3 s long moving time window with an overlap of 70%.

Even though this group contains 11 examples, it is difficult to identify the source of the coherent noise since the spikes in this group are always either preceded or followed by another package of noise without any regular pattern. Events in this group are strongly visible on the east and north components. Since they are not regular but distributed randomly in the dataset and are short-period noise, we cannot identify the source.

To conclude, we investigated the coherent noise signals and compared the recordings from different sensors. As a side product, we tried to speculate the potential sources of coherent noise signals. Spectrograms of the waveforms in each group of coherent noise were consistent for all sensors. Two out of five noise signal groups, namely the long-lasting and high frequency spike-shaped noise, were not visible on the vertical components. This suggests that the source radiated a wavefield that is dominated by SV-type waves. In general, when coherent noise is present, each sensor detects it and shows a similar frequency content in the spectrograms. Especially the north and vertical components of BS1 and BS2 show higher amplitudes when compared to the other sensors.

#### 4.2.2. Correlation

Correlation studies have been frequently used to detect seismic events with lower magnitudes [26,27,28], to calculate the locations of seismic events (e.g., [29,30,31]), and to investigate coherent noise in several ambient noise studies, (e.g., [32,33,34]). Here, we calculated the correlation between all three components of all rotational sensors to investigate the similarity. Here, we used a 3–9 Hz bandpass filter. We used sliding time windows of 5 s, 10 s, 20 s, 1 min, 5 min, and 10 min duration to choose the optimal sliding time window. The correlation results are best when using a 5 s moving time window, which we use with 50% overlap. We present the representative correlation results between east components (HJ1/HHE) of rotational–rotational, rotational–translational, and translational–translational sensors for doy 240, i.e., 28 August, Wednesday (Figure 11).

Due to the sensors’ self-noise levels, we expected to estimate positive but not strong correlation results. The mean correlation between XB101 and XB102 was evaluated as 0.642 ± 0.033. The mean of the correlation between one translational (HRZ3) BS one rotational (XB101) sensor was 0.641 ± 0.034. Furthermore, the mean of the correlation for two translational sensors (HRZ1–HRZ3) was 0.665 ± 0.044. Coherent noise signals lead to an increase in correlation between two sensors. Since the translational instruments are sensitive enough to record ambient noise, a higher correlation might be expected for the two translational sensors. We assumed that the translational data record microseism well and, thus, provide high correlation values even outside the time windows with coherent noise signal. The rotational energy of the microseism is too small to be recorded by the rotational sensors. Hence, the correlation between rotational sensors is lower when no coherent noise source is present.

To sum up, correlation of the waveforms from rotational and translational sensors over a 24 h time frame is estimated in this section. The results show similar values and a positive correlation in time. When the correlation of each sensor is investigated within a specific time window that includes a coherent noise (e.g., in Section 4.3), it shows high correlation results in comparison to the 24 h long correlation.

### 4.3. Analysis of the Dettingen Earthquake Using Rotational Sensors and Seismometers

Within the few days of the huddle test, 12 earthquakes from local to regional distance ranges are extracted from several catalogues (Table 2). Five of them with magnitudes between M${}_{L}$ 3.4 and M${}_{w}$ 4.8 and distances 168 km to 1930 km could be recorded on the translational sensors. Only one earthquake was also recorded by the rotational instruments. It has a magnitude of $M_{L}$ 3.4 and occurred on Thursday, 29 August (doy 241) close to Dettingen, Germany (distance ∼168 km). In the following, we used this recording for further tests of the signal coherence on the different sensors and to determine the BAz and apparent phase velocity.

#### 4.3.1. Signal Coherence

As an addendum to the coherence analysis on the noise data in the previous section, here, we use the signal of the Dettingen earthquake to again analyze the data coherence between the rotational sensors. We applied the cross-correlation method described in Hadziioannou et al. [21]. We used a two minute long data segment starting at the origin time of the event and filtered it between 3 and 9 Hz. The data are analyzed in a 1 s window with 50% overlap. In each window, a correlation coefficient is calculated. For all time windows with correlation coefficients exceeding 0.75, we calculated the average back azimuth value for each trace. As an example, Figure 12 shows the result for the sensors XBLUE and XB101. For the other sensor comparisons, the results are summarized in Table 3.

Having different internal Butterworth low-pass filters with nonlinear phase responses, the analyzed waveforms were shifted in time, and as a result, all sensors show high correlation values between 80 and 92 %. For comparison, the correlation between the translational sensors HRZ1 and HRZ3 resulted in correlation values of, on average, 96/86/97 % for the HHZ/HHN/HHE component. It is interesting to mention that the correlation values on the translational data form a plateau over the entire time of the analyzed trace around the mentioned values. In contrast, the correlation values on the rotational sensors includes ups and downs (see Figure 12). The highest correlation appears at the S-wave with a smaller increase shortly behind the P-wave.

#### 4.3.2. Determining the Back Azimuth

The real back azimuth (BAz) according to the coordinates from the catalogue is 254.5°. We applied the cross-correlation method [21] (see Section 3) on the recordings of all possible combinations of rotational sensors and seismometers. According to the spectrogram of the vertical rotation trace, the energy of the event is concentrated in the frequency band of 3 Hz to 10 Hz, which we therefore use for the analysis. To analyze microseism, Hadziioannou et al. [21] chose a sliding time window that is fixed at the central frequency of the processed frequency range. In our case, that would be a window of about 0.3 s, which is too small. For the processing of the global database, Salvermoser et al. [35] applied a sliding time window of 3 s for events at a distance range of 0° to 3°. In our case, a sliding time window of 1 s proved to be optimal. The resulting plot from XBLUE and HRZ1 is shown in Figure 13.

The highest coherence appears at the onset of the S-wave and in the S-wave coda, indicating sensitivity for Love-wave energy though they are not clearly visible in the waveform data. Here, the determined BAz and the true BAz fit perfectly. Shortly after the P- and S-wave onsets, we observe high cross-correlation coefficients. We assume that this short time shift is caused by scattered energy within the medium. The determined phase velocities range from 0.3 to 1.0 km/s (Table 4). These values seem realistic given that the applied method is sensitive to SH and Love-wave energy. Wassermann et al. [36] derived an S-wave velocity of about 0.5 km/s for the upper 50 m from array-derived rotation (ADR) analysis as well as from spatial autocorrelation (SPAC).

In sliding windows with correlation coefficients higher than 0.65, we calculated the average BAz and phase velocity value and its standard deviation. We repeated the analysis for all five rotational sensors in combination with two translational sensors and show the mean ± standard deviation results for the BAz determination and phase velocity estimates in Table 4.

The mean BAz and phase velocity and their standard deviation values were calculated based on timestamp-corrected traces (see explanation in Section 3). The mean and standard deviation of the BAz and phase velocity results from all sensors are calculated with a cross-correlation threshold of 0.65. Since this BAz estimation method is sensitive to SH-type waves, high correlation values near the arrival of the P-wave are likely phase conversion or scattering. We observed that the calculated BAz values for the P-waves decrease the mean values and increase the standard deviation stated in Table 4. We chose to exclude the P-wave energy since our sensors are directly sensitive to S-wave energy and the correlation values are high.

Altogether, XBLUE, XB101, and XB102 yielded BAz values with smaller standard deviations, which are realistic and comparable to the real BAz values. These sensors yield maximum phase velocities around 1 km/s and mean phase velocities in the range of 0.49 to 0.63 km/s. For BS1 and BS2, time-shifts were applied that give maximum correlation between rotational and translational recordings. Following this correction, BS1 and BS2 show comparable BAz results but still yield lower phase velocities. The phase velocity values with the highest correlation coefficients were reached in the coda of the earthquake for BS1 and BS2 and at the S-wave arrival for the other sensors.

## 5. Conclusions

Rotational motions have attracted scientific attention in the field of seismology in recent years. Due to the lack of sensitive and reliable sensors, rotational motions could not be properly studied in the past. However, with recent developments, portable sensors have been used to directly measure rotational motions for seismological purposes. The huddle test presented here provides an opportunity to test the sensor performance and reliability of these newly developed rotational instruments. Thus, in this paper, the data from five rotational and two translational co-located instruments are presented. Recordings from different sensors were corrected in time before investigating coherency and back azimuth due to differences in the reference clock, time synchronization, and causal low-pass filtering prior to digital sampling decimation used in each sensor. The self-noise levels of each sensor are shown to be consistent and stable between 0.5 to 40 Hz. Coherent noises are separated into five groups. Events within these groups have similar frequency content, duration, and shape and are recorded by all sensors. The BS1 and BS2 sensors showed higher amplitudes in comparison to the XBLUE, XB101, and XB102 sensors. The correlation between all sensor pairs is positive and stable, and when coherent signals are recorded, the correlation increased. Signal coherence over the course of an earthquake was strongly positive for all rotational sensors. Back azimuth calculations using the vertical rotational rate measurement of one rotational sensor and transverse acceleration of one seismometer were all consistent with the back azimuth based on the earthquake catalog. In conclusion, despite being new developments, multiple blueSeis-3A rotational sensors yield consistent results within the aforementioned band of interest and are therefore reliable instruments to be used in the field.

## Figures and Tables

**Figure 1 sensors-21-03170-f001:**
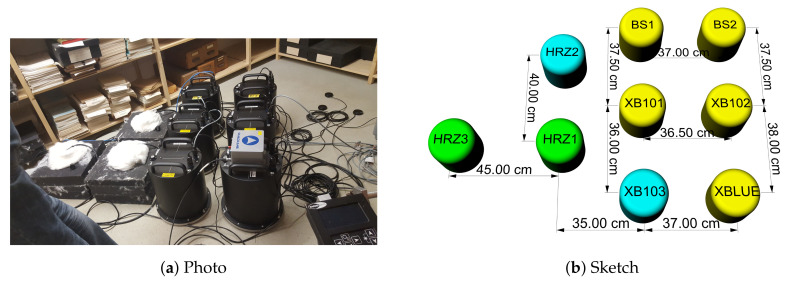
Sensor setup for the huddle test in the basement of the Observatory in Fürstenfeldbruck, Germany. (**a**) 3 Trillium Horizon 120s Nanometrics seismometers in left are isolated with black foam rubber and white cotton on top. 6 blueSeis-3A instruments in the middle, are linked to GPS devices in top right corner. The Quadrans is on top of XBLUE used for orientation. (**b**) Sketch of the experiment setup where 2 recording Trillium Horizon 120s Nanometrics seismometers (green HRZ1 and HRZ3), 5 recording blueSeis-3A rotational seismometers (yellow BS1, BS2, XB101, XB102 and XBLUE) and 2 not working instruments (cyan XB103 and HRZ2) are shown.

**Figure 2 sensors-21-03170-f002:**
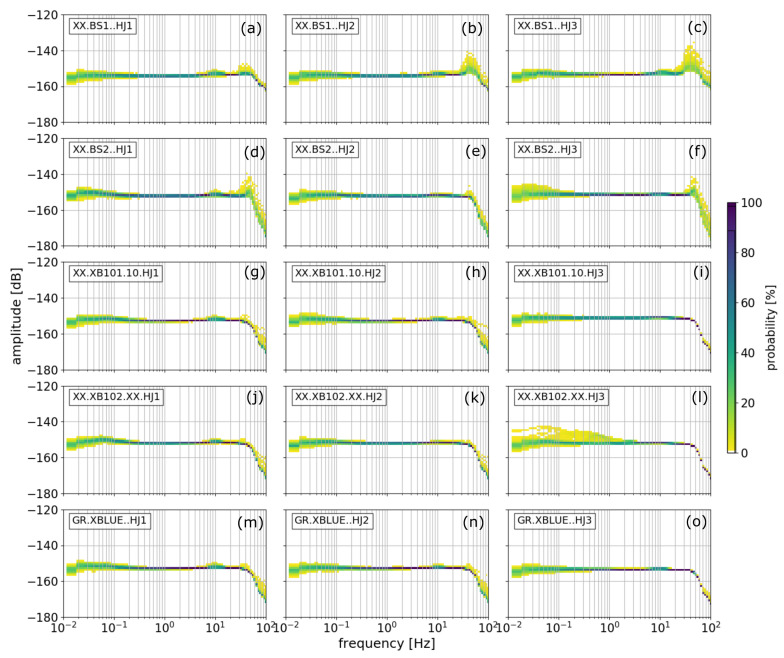
Probabilistic power spectral densities for all three components of all five rotational sensors for 29 August, Thursday. Components HJ1 (**a**,**d**,**g**,**j**,**m**) and HJ2 (**b**,**e**,**h**,**k**,**n**) correspond to the East and North component while component HJ3 (**c**,**f**,**i**,**l**,**o**) corresponds to the vertical component, respectively.

**Figure 3 sensors-21-03170-f003:**
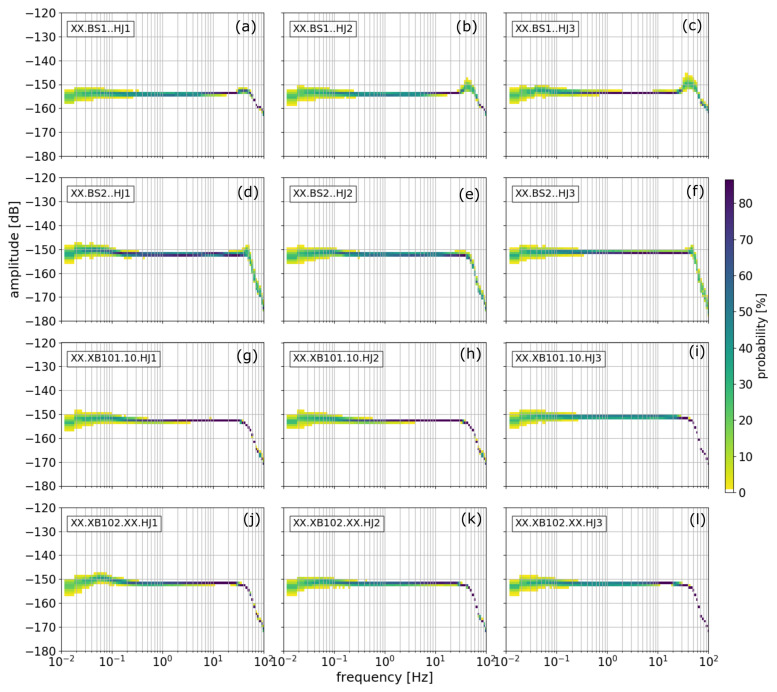
Probabilistic power spectral densities for all three components of all five instruments for 1 September, Sunday. Panel content like in Figure 2.

**Figure 4 sensors-21-03170-f004:**
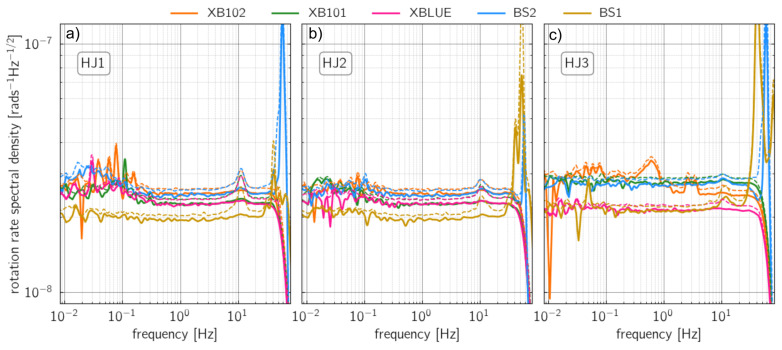
Self-noise power spectral densities (PSD) of XB102 (Orange), XB101 (Green), XBLUE (Pink), BS1 (Yellow) and BS2 (Blue) for the East (**a**), North (**b**) and vertical components (**c**). The dashed line represents the simple single sensor PSD. The solid lines show the results from the 3-station method after Sleeman et al. [12].

**Figure 5 sensors-21-03170-f005:**
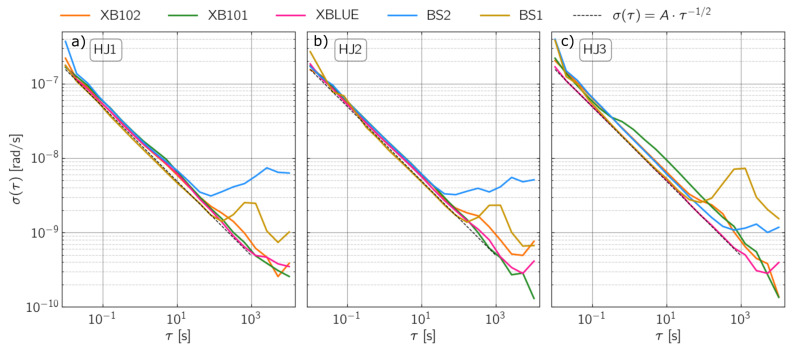
Allan Deviations for all sensors and all three components. (**a**) represents East, (**b**) North and (**c**) vertical components, respectively. The dashed line represents the white noise.

**Figure 6 sensors-21-03170-f006:**
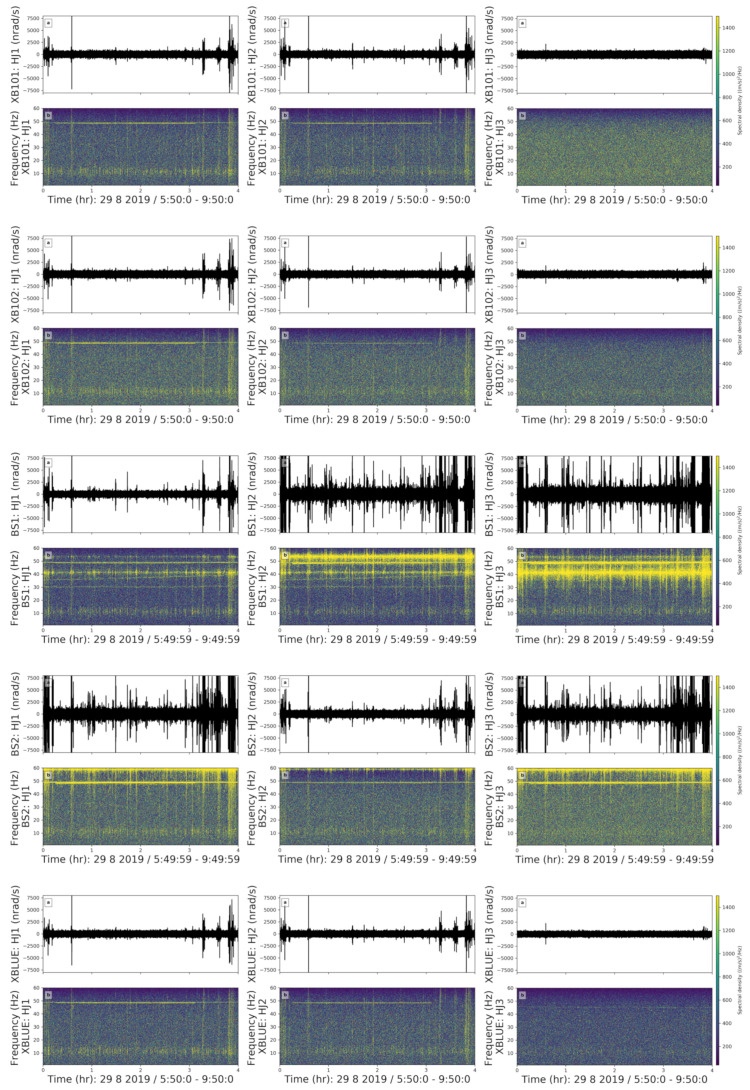
Long-lasting noise source. (**a**) Waveforms and (**b**) Spectrograms of the East, North and vertical components (left, middle, right column) of the XB101, XB102, BS1, BS2, XBLUE sensors from top to bottom for 4 h on doy: 241, i.e., 29 August, Thursday, respectively. Yellow and blue colours indicate higher and lower spectral densities, respectively.

**Figure 7 sensors-21-03170-f007:**
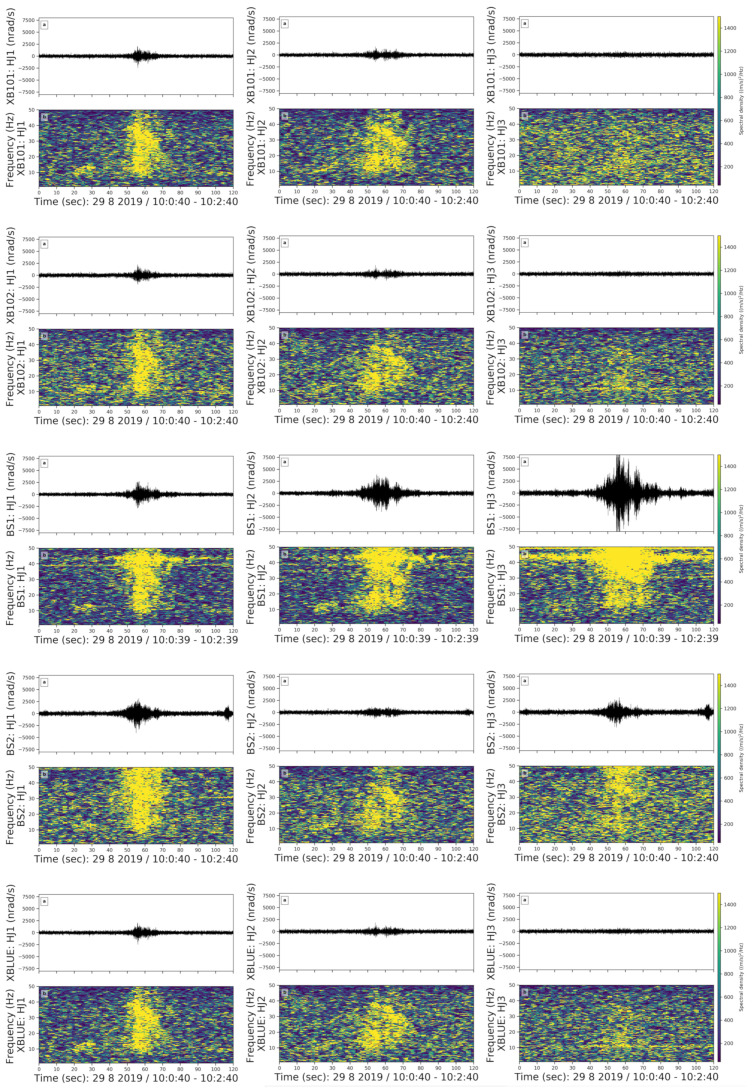
Emergent burst-like noise in a 2 min long time window on doy 241, i.e., 29 August, Thursday. The subfigure structure, content and colors are the same as in Figure 6.

**Figure 8 sensors-21-03170-f008:**
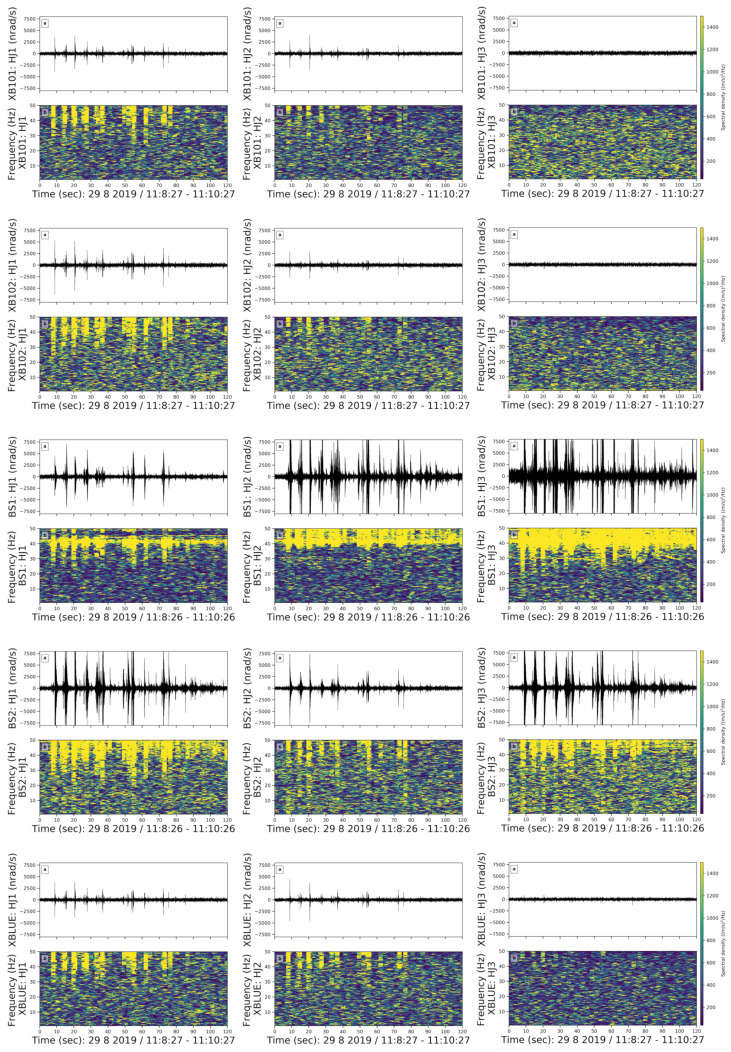
High-frequency spike-shaped noise in a 2 min long time window on doy 241, i.e., 29 August, Thursday. The subfigures and colors are the same as in Figure 6.

**Figure 9 sensors-21-03170-f009:**
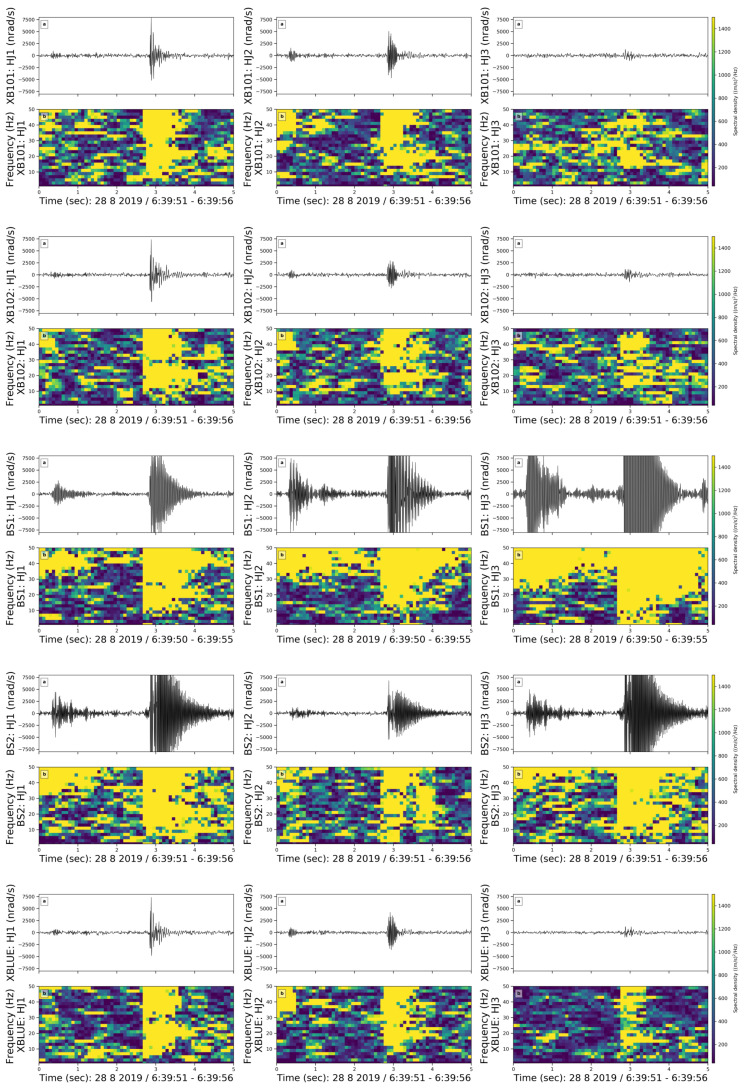
Very short noise in a 5 s long time window on doy 240, i.e., 28 August, Wednesday. The subfigures and colors are the same as in Figure 6.

**Figure 10 sensors-21-03170-f010:**
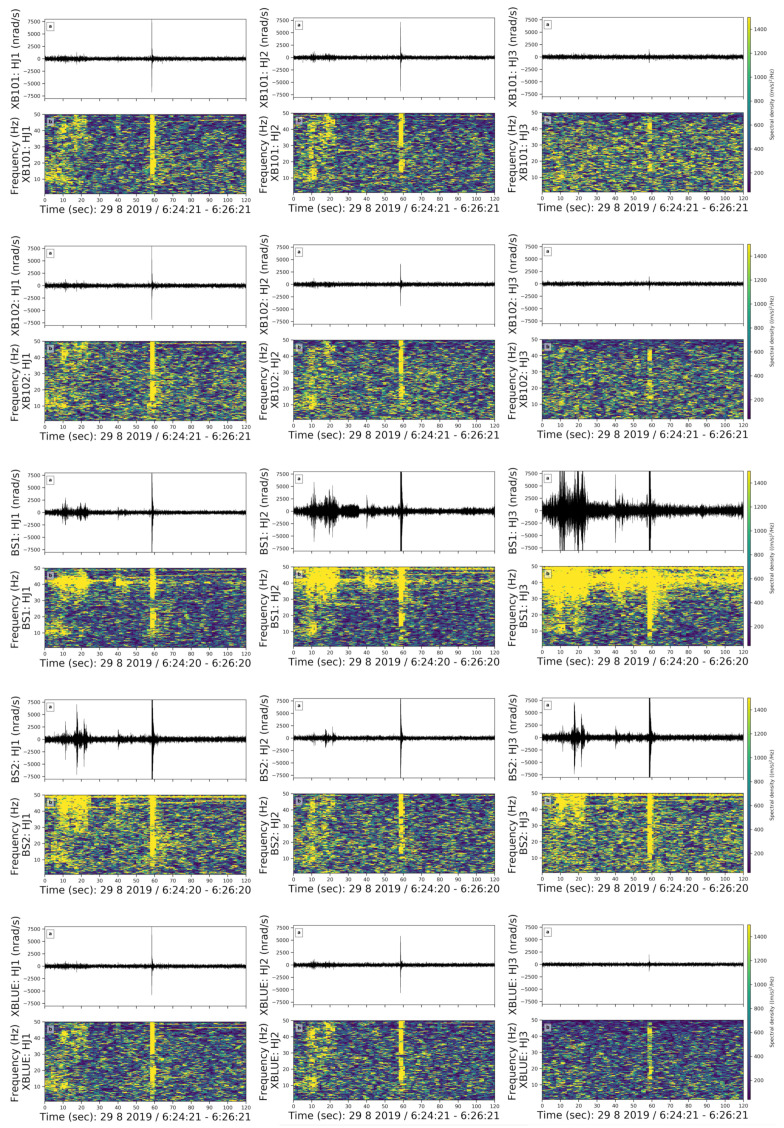
Spike-shaped noise in a broad frequency range in a 2 min long time window on doy 241, i.e., 29 August, Thursday. The subfigures and colors are the same as in Figure 6.

**Figure 11 sensors-21-03170-f011:**
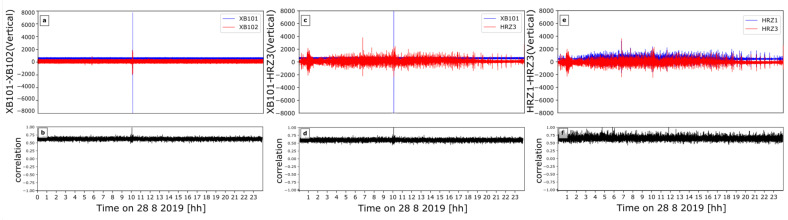
(**a**) Seismograms of XB101 (Blue)—XB102 (Red), (**c**) XB101 (Blue)—HRZ3 (Red), (**e**) HRZ1 (Blue)—HRZ3 (Red) from left to right respectively. Correlation (Black) between (**b**) XB101-XB102, (**d**) XB101-HRZ3, (**f**) HRZ1-HRZ3 sensors for doy 240, i.e., 28 August, Wednesday.

**Figure 12 sensors-21-03170-f012:**
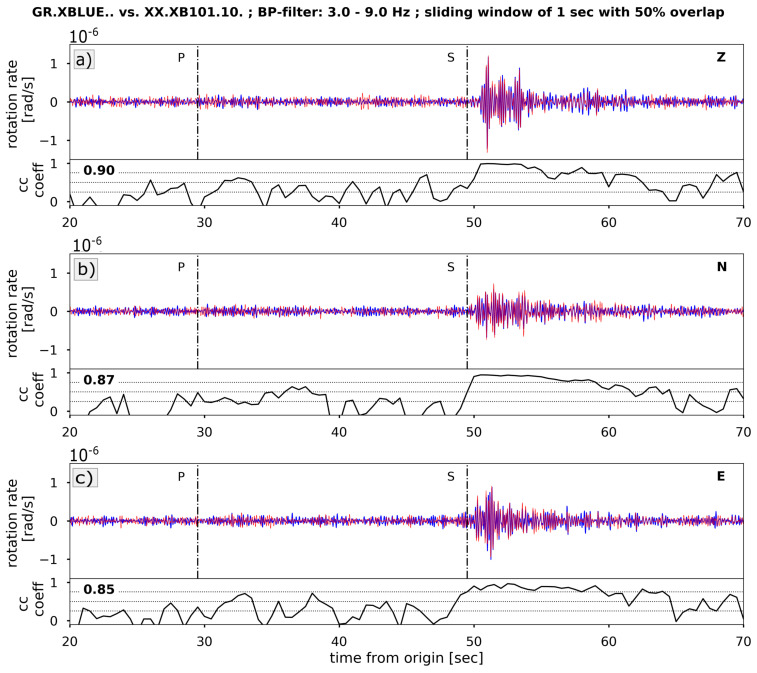
Component-wise cross-correlation between rotational sensors XBLUE (blue) and XB101 (red). Data are filtered between 3 and 9 Hz and analysed in a sliding window of 1 s length with 50% overlap. Vertical dotted lines mark the onsets of the P- and S-wave. (**a**–**c**) show the correlation coefficient over time for each component. The average value for all coefficients greater than 0.75 is given in the upper left corner. For other sensor combinations results are summarised in Table 3.

**Figure 13 sensors-21-03170-f013:**
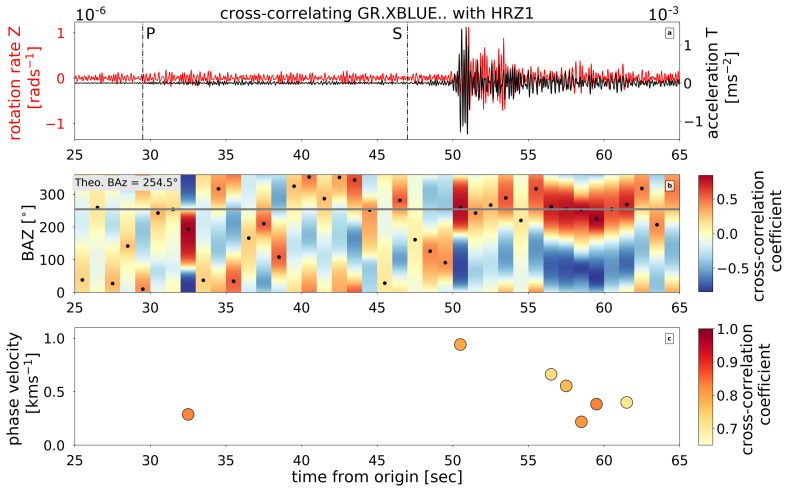
(**a**) Cross-correlation of the vertical rotation rate (sensor XBLUE) and the transversal acceleration (sensor HRZ1). Data are analysed in a sliding time window of 1 s with no overlap. (**b**) To obtain the transverse component, North and East components are rotated by all possible BAz between 0 and 360° in increments of 5 ° in each sliding window. CC coefficients are calculated for each BAz in each sliding window. The horizontal grey line marks the real BAz of 254.5°. (**c**) For windows and BAz with cc coefficient > 0.65 apparent phase velocity is calculated according to Figure 7.

**Table 1 sensors-21-03170-t001:** Parameters of the installed instruments.

	Sensor	Name	Serial Number	Position	Start UTC	Stop UTC	Sampling Rate
Rotational		XB103	19049	front-left	(off)	-	200 Hz
		XBLUE	19023	front-right	239 11:20	243 19:02	
	iXblue	XB101	19046	middle-left	239 10:51	245 07:09	
	blueSeis-3A	XB102	19009	middle-right	239 10:51	245 07:09	
		BS1	n.a.	back-left	239 00:00	245 07:08	
		BS2	18202	back-right	239 00:00	245 07:11	
Translational	Nanometrics	HRZ3		front-left	238 10:34	245 09:43	200 Hz
	Trillium	HRZ1		front-right	238 10:36	245 09:41	
	Horizon 120 s	HRZ2		back	238 10:34	245 09:42	

**Table 2 sensors-21-03170-t002:** Earthquakes within the time range of the huddle test. dist gives the distance of the earthquake to Fürstenfeldbruck. T/R states whether the event was recorded on translational and/or rotational sensors.

Date	Time (UTC)	Lat (°)	Lon (°)	Place	Magnitude	Dist (km)	T/R	Source
08-27	10:10:12	49.003 N	11.427 E	Beilngries, D	ML 1.9	94	n/n	BGR
08-28	11:58:42	35.65 N	27.56 E	Dodecanese, Greece	Mw 4.8	1930	y/n	EMSC
08-29	00:50:11	51.63 N	16.19 E	Poland	Mb 4.1	523	y/n	IRIS Wilber 3
08-29	12:27:13	50.028 N	12.233 E	Konnersreuth, D	ML 0.4	219	n/n	EQ Serv. Bavaria
08-29	14:22:48	47.740 N	9.108 E	Dettingen, D	ML 3.4	168	y/y	BGR
08-30	05:57:52	23.766 N	45.446 W	Atlantic Ridge	Mw 5.3	5629	n/n	USGS
08-30	07:00:04	45.73 N	26.64 E	Romania	Mb 4.5	1198	n/n	IRIS Wilber 3
08-30	13:01:36	44.63 N	18.54 E	Bosnia/Herzegovina	Mb 4.8	682	y/n	IRIS Wilber 3
08-30	17:21:05	37.53 N	26.78 E	Dodecanese, Greece	Mw 4.5	1726	n/n	EMSC
09-01	00:02:39	42.81 N	13.10 E	Central Italy	Mw 4.1	612	y/n	IRIS Wilber 3
09-01	20:34:04	53.99 N	35.53 W	Reykjanes Ridge	Mb 4.4	3281	n/n	IRIS Wilber 3
09-02	08:39:45	49.383 N	10.156 E	Taubertal, D	ML 1.9	159	n/n	BGR

**Table 3 sensors-21-03170-t003:** Maximum CC coefficients for each component (HJ3/HJ2/HJ1) and each sensor in % in a 50 s long time window containing the Dettingen earthquake (average for coefficients > 0.75). An example for the comparison of the sensors XBLUE and XB101 is shown in Figure 12.

	XB101	XB102	BS1	BS2
XBLUE	90/87/85	92/89/88	82/86/86	80/84/79
XB101	-	89/86/84	87/89/84	81/81/79
XB102	-	-	83/85/86	80/80/81
BS1	-	-	-	75/82/79

**Table 4 sensors-21-03170-t004:** BAz estimates in ° and phase velocity estimates in km/s for all possible combinations of five rotational sensors and two seismometers. Mean values and standard deviation are calculated for all time windows with CC coefficients of >0.65 for phase velocity and BAz. The real BAz is 254.5°.

CC Method		XB101	XB102	BS1	BS2	XBLUE
HRZ1	BAz (°)	264.0 ± 15.95	258.28 ± 13.77	235.0 ± 26.8	274.25 ± 16.39	253.83 ± 13.88
	Phase velocity (km/s)	0.50 ± 0.23	0.63 ± 0.19	0.32 ± 0.07	0.42 ± 0.21	0.55 ± 0.22
HRZ3	BAz (°)	263.25 ± 15.83	257.71 ± 13.43	234.6 ± 26.45	273.75 ± 16.11	253.16 ± 13.60
	Phase velocity (km/s)	0.49 ± 0.23	0.64 ± 0.19	0.32 ± 0.07	0.42 ± 0.21	0.55 ± 0.22

## Data Availability

Seismological data is available through GEOFON (doi:10.14470/K87564381089).

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
