# Peer review of "Performance Test of the Rotational Sensor blueSeis-3A in a Huddle Test in Fürstenfeldbruck"

_sensors, 2021, doi:10.3390/s21093170_

Round 1

Reviewer 1 Report

The paper presents the results from a passive field experiment that includes 6 rotational and 3 translational sensors. The objective under investigation consists of assessing the performance of multiple blueSeis-3A sensors in combination with translational seismometers. In this regard, a huddle test was performed at the Geophysical Observatory in Fürstenfeldbruck, Germany, as a collaborative work of Ludwig-Maximilians University of Munich (LMU), University of Potsdam (UP) and the Federal Institute for Geosciences and Natural Resources (BGR).   The topic is interesting and relevant. The methodology pertinent and well explained. However, the organization of the paper needs to be reconsidered and improved. English also needs to be checked and more fluent. Some points need to be developed and detailed before the final submission:   In the introduction, it is important to clearly specify the novelties and the scope of the study. Is it a case study? A proposal of a new methodology? Please clarify.     Section 3. The title should be methodology   Section 3.3 The authors need to extend the explanation of this part in order to improve the understandability of the whole paper.   Please rewrite this sentence: "Additionally, the translational data is instrument corrected."   Section 4 needs to be splitted at least in two parts that consider the two methods presented in section 4.1.1. In particular, section 4.2.1 discusses the quality and reliability of the data over coherent noise and correlation study and cannot be strictly considered "results" Section 4.3 is the real "results" section, where the authors present the results from coherency analysis and back azimuth calculations     Conclusion Please rewrite these sentences in order to clarify the novel contributions:   "Rotational motions attract more scientific attention in the field of seismology in recent years." Extend a little more this idea.   "A huddle test provides an opportunity to test the sensor performance of the relatively newly developed rotational instruments." Clarify better how this test consists of.        

Author Response

Dear reviewer,

We appreciate very much your review of our submitted manuscript. We read thoroughly your comments and made changes and corrections to the manuscript. We hope our answers and changes are sufficient to make our article suitable for publication soon. Your comments and suggestions helped to improve quality and clarity of the paper.

Our replies (-) to your comments are given in the attachment.

Thank you again, best regards

Gizem Izgi

Reviewer 2 Report

Review of Performance Test of the Rotational Sensor blueSeis-3A in a Huddle Test in Fürstenfeldbruck.

The authors present results of a huddle test with 6 rotational sensors and 3 translational sensors.  The paper is well written and the results are interesting and communicated well.  I think this is a valuable contribution to seismology as it helps provide documentation on a a small sample set of instruments that have been newly developed.  I have just a few comments/corrections below.  I look forward to seeing the published version of this paper.

If you used obsPy to create the figures you should reference one (or more of their papers) so they get credit.  Is this data available somewhere?  If so you might put that in the paper.

Line 23 and 24: This sentence has some issues with it.  Please double check it.  You have displacement/velocity/acceleration, respectively.  This seems off.

Line 68: Reword this as it might not be sufficiently close, but you are assuming it is.  

Line 74: This is an interesting result.  Our current orientation devices are better than our ability to do the orientation.

Lines 87 and 88: You want that the single instrument method simply estimates the background noise + the instrument noise.  So that in high-noise environments you might not get a true reflection of the self-noise.  The PPSD method estimates the statistical spread of this noise.

Line 89: remove very well established.  Perhaps reword to “they developed a method that has become widely established.”

Line 90: I believe Sleeman developed it initially for digitizers, double check the reference.  I could be wrong.

Line 94: You want that this is a function of frequency.  

Lines 94 through 100: Could you expand this a bit more.  This would be hard for someone not familiar with to read.  Or you could say you follow the conventions of paper blah blah and then remove some of this.

Line 102: You have a typo, it should be Buland.

Lines 140 and 141: Wouldn’t anti-alias filters only change the high-frequency portion of the data?

Line 161: It is coherent, but it could be that it is a resonance in the sensor.  So you are getting a resonant peak and removing it using coherence.  So it might not be seismic.

Line 247: Can you really have noise signal.  Wouldn’t it be either noise or signal?  

Table 4: Do you think your azimuthal errors are coming from scattered energy?  Or what do you think the issue is?  These do not strike me as great uncertainties.

Line 340: Please reword this to be more natural.

Line 346: Change proven to shown. 

Line 353: Do you want to say within your band of interest?  Your study was isolated to a relatively narrow band.  

Author Response

(The authors gave the same response as above.)
